# Marf- and Opa1-Dependent Formation of Mitochondrial Network Structure Is Required for Cell Growth and Subsequent Meiosis in *Drosophila* Males

**DOI:** 10.3390/ijms26209991

**Published:** 2025-10-14

**Authors:** Tatsuru Matsuo, Mitsuki Yamanaka, Yoshihiro H. Inoue

**Affiliations:** Biomedical Research Center, Kyoto Institute of Technology, Matsugasaki, Sakyo-ku, Kyoto 606-0962, Japan; tatsuru27@gmail.com (T.M.); kitstbushi2012@gmail.com (M.Y.)

**Keywords:** mitochondrial dynamics, *Drosophila*, meiosis, spermatogenesis, Cyclin B, nebenkern

## Abstract

Mitochondria are dynamic organelles that undergo repeated fusion and fission. We studied how the distribution and shape of mitochondria change during *Drosophila* spermatogenesis and whether factors that regulate their dynamics are necessary for these changes. Unlike the shortened mitochondria seen in mitosis, an interconnected network of elongated mitochondria forms before meiosis and is maintained during meiotic divisions. Mitochondria are evenly divided into daughter cells, relying on microtubules and F-actin. To explore the role of mitochondrial network structure in cell growth and meiosis, we depleted the mitochondrial fusion factors Opa1 and Marf and the morphology proteins Letm1 and EndoB in spermatocytes. This knockdown led to inhibited cell growth and failed meiosis. As a result, the spermatocytes differentiated into spermatids without completing meiosis. The knockdown also inhibited the cytoplasmic and nuclear accumulation of Cyclin B before meiosis, and Cdk1 was not fully activated at the onset of meiosis. Notably, ectopic overexpression of Cyclin B partially rescued the failure of meiosis. Many spermatids from the spermatocytes subjected to the knockdowns contained multiple smaller nuclei and abnormally shaped Nebenkerns. These findings suggest that mitochondrial network structure, maintained by fusion and morphology factors, is essential for meiosis progression and Nebenkern formation in *Drosophila* spermatogenesis.

## 1. Introduction

Although mitochondria have been depicted as granular, capsule-like structures based on past transmission electron microscope images, they also come in various other morphological forms, including interconnected structures made of elongated mitochondria called network structures [1,2,3,4]. When oxidative damage is generated in mitochondria, fusion between neighboring mitochondria allows damaged proteins and DNA to be replaced with normal versions [5,6]. Damaged mitochondria also undergo fission, and the smaller parts with the damaged components are eliminated via mitophagy. Even during normal cell proliferation, the morphology changes depending on the phase of the cell cycle [7]. The elongated structures formed via the fusion of multiple mitochondria in mammalian cells transform into granular structures just before entering M-phase, and are reassembled at the end of M-phase [1]. The ability of mitochondria to adopt different shapes is postulated to be related to their dynamic nature, involving repeated fusion and fission.

Three GTPases which are essential for mitochondrial fusion and fission were originally identified in yeast [8]: the cytoplasmic dynamin-related GTPase Drp1 plays a key role in mitochondrial fission, Mfn1/2 plays a role in the outer- membrane fusion of mitochondria, and Opa1 plays a role in inner-membrane fusion of mitochondria. Their orthologs have been identified in mammals and *Drosophila* [9,10,11]. If these factors do not function properly, the balance between fusion and fission is shifted, which can impact mitochondrial morphogenesis and function. In addition, LETM1 is a mitochondrial inner-membrane protein that is required in processes related to mitochondrial morphology and cristae structures [12]. Disrupting mammalian Endophilin B1, which is located on the outer membrane of mitochondria, also causes abnormal elongation of the outer mitochondrial membrane and disturbs the balance between fusion and fission [13]. Here, we refer to these two membrane proteins as mitochondrial morphology proteins and study them together with the fusion and fission factors. Mitochondrial morphology is closely associated with important biological phenomena, including the cell cycle, apoptosis, and developmental events [14,15,16,17].

In *Drosophila*, mitochondria undergo remarkable morphological changes during spermatogenesis [18,19,20,21]. The early-stage spermatocytes (at the S2b stage) are characterized by a polar distribution of the nucleus and an adjacent mitochondrial cluster. Thereafter, in the S3–5 stage, mitochondria are dispersed throughout the cytoplasm. By the end of the growth phase (S6 stage), spermatocytes reach their maximum size and mitochondria appear to make clusters again. After the growth phase, they undergo two consecutive meiotic divisions. Mitochondria are equally partitioned into the two daughter cells during meiotic divisions while being closely associated with microtubule structures specifically constructed for male meiosis [18,19,20]. After the second meiosis, 64 spermatids formed simultaneously. Mitochondria in a post-meiotic cell, designated a spermatid, construct a large and spherical aggregate known as a Nebenkern next to the nucleus. Mitochondria in a post-meiotic cell, known as a spermatid, construct a large and spherical aggregate known as a Nebenkern next to the nucleus. *Drosophila* has two orthologues of the mammalian mitochondrial outer-membrane fusion factors: Mitochondrial assembly regulatory factor (Marf) and Fuzzy onion (Fzo). The *fzo* mutants form multiple small Nebenkerns [7]. Similar abnormalities in Nebenkern were reported in mutant spermatocytes for *Opa1* [22]. The cells lacking *Drp1*, encoding a mitochondrial fission factor, also display a similar Nebenkern phenotype [23]. However, the fine morphology and distribution of mitochondria before and during meiosis, as well as the regulatory factors involved in these processes, remain to be investigated.

In this study, we focused on the morphological changes mitochondria undergo during *Drosophila* spermatogenesis, especially before and during meiosis. We investigated whether the mitochondrial fusion factors, Marf and Opa1, are required for the formation of a network structure consisting of the elongated mitochondria. We investigated whether the morphology proteins Letm1 and EndoB are also required for the mitochondrial morphological changes observed. Moreover, we investigated whether the mitochondrial network structure constructed via these gene products plays an indispensable role in male meiosis. We also addressed the mechanism underlying the fact that the knockdown of the fusion genes essential for the network structure inhibits meiosis. Since mitochondria are essential organelles in spermatogenesis, not only in *Drosophila* but also in mammals, our findings may provide important insights into the roles of mitochondrial dynamics in spermatogenesis.

## 2. Results

### 2.1. Dynamics of Mitochondrial Morphology in Drosophila Early Spermatogenesis and Meiotic Division

During the mitosis of a spermatogonium arising from a germline stem cell, mitochondria were distributed in the cytoplasm of daughter cells in a dot-like morphology (as shown by the arrowheads in Figure 1(A2,B2)). During interphase, when an aster was localized on the nuclear membrane, mitochondria diffused into the cytoplasm in granular and interconnected forms (as shown by the arrow in Figure 1(C2)). Sixteen spermatogonia generated via four rounds of mitosis formed a single cell unit—namely, a cyst—and underwent cell growth synchronously. During an early stage (S2b) of the cell growth phase, elongated mitochondria accumulated next to the nucleus (as shown by the arrows in Figure 1(D1)—see the super-resolution microscopic images in Appendix A(A2,A3)). As the development of spermatocytes progressed from the S2b to the S3–S5 stages, the mitochondrial network transformed, adopting a granular and shortened form (Figure 1(E1), see Appendix A(B2,B3)). By the mature stage (S6), just before the onset of meiosis, a network structure consisting of elongated mitochondria was constructed (as shown by the arrows in Appendix A(D2,D3) and the arrow in the inset of Figure 1(F2)). During meiosis I, this network structure was maintained and distributed around aster microtubules (Figure 1(G1)). By metaphase I, mitochondria had migrated toward the plus ends and assembled along the vicinity of the plasma membrane toward the cell equator between the two asters (Figure 1(H1)—see *t* = 10–40 min in the live analysis in Appendix A). At this stage, the structure of the mitochondrial network was not fragmented or distributed, and it retained its original structure (see Appendix A(E1,F1)). In late anaphase I, as central spindle microtubules formed between daughter nuclei, mitochondria shifted from being near the plasma membrane to the center, where these microtubules are located (as shown by the arrows in Figure 1(I2)). By the end of anaphase I, mitochondria were distributed in a ribbon-like shape on the central spindle (Figure 1(J2)). As the spindle structure disintegrated during telophase I, mitochondria continued to exhibit this shape distribution (Figure 1(K1)). Furthermore, we used transmission electron microscopy to confirm the presence of elongated mitochondria during anaphase I to telophase I (Appendix A(G1,G2,H1,H2)). After the cytokinesis of the first meiotic division, mitochondria appeared to diffuse into the cytoplasm as the microtubule structure disintegrated. At this stage, mitochondria were dispersed into the cytoplasm while maintaining an elongated network structure (inset in Figure 1(L2)). The mitochondria were distributed in the same manner in the second meiosis. After the completion of meiosis, the mitochondria formed a single aggregate, known as a Nebenkern (Figure 1(M1)).

Next, we examined whether mitochondria maintain a membrane potential essential for ATP synthesis during meiotic division. Active mitochondria stained with Mito Tracker have an active electron-transfer system and are capable of ATP synthesis. Mitochondria were distributed to daughter cells through meiosis I (Appendix A), with their ATP-synthesizing ability maintained.

### 2.2. The Impact of Microtubule and F-Actin Depolymerization on Mitochondrial Network Formation and Distribution

We next investigated whether the changes in mitochondrial distribution depend on microtubules or the F-actin cytoskeleton. Even after depolymerization of microtubules, mitochondria continued to assemble next to the nucleus in the early stage (S2b) of the cell growth phase (Figure 2(A1)), as in the untreated cells. Mitochondria were observed in the same elongated network structure (as shown by the arrows in Figure 2(A2) and Figure 1(D2)). In the untreated cells, this network structure was constructed over the period from the mature stage (S6) to prophase I (Figure 1(F1)).

Despite the absence of microtubules (Figure 2(B3)), a similar mitochondrial structure was observed in the spermatocytes (Figure 2(B2)) (102 cells/176 S6-like cells), suggesting that microtubules are not essential in the formation of the network structure by mitochondria. Despite the aster microtubules being disintegrated in the prometaphase I cells (Figure 2(D3)), mitochondria left the position where microtubules were present (Figure 2(C2–E2)) and maintained their network structure (32 cells/50 cells). The mitochondrial structure, which was oriented toward the spindle poles in the untreated cells (Figure 1(I2,J2)), lost its orientation after the treatment (as shown by the arrows in Figure 2(E1,F1)) (35 cells/46 cells). In contrast, the ribbon-like shape of the mitochondrial structure (Figure 1(J1,K1)) was preserved in meiotic cells lacking microtubules (Figure 2(F1)). Mitochondria accumulated between the separated homologous chromosomes, even if the central spindle structure had disintegrated. The mitochondria then appeared to be distributed throughout the cytoplasm during cytokinesis, as seen in the untreated cells (Figure 2(G1)).

Next, we examined whether F-actin is also required for mitochondrial morphology and distribution in pre-meiotic and meiotic cells. Both mitochondria that had accumulated in the cytoplasm of the polar spermatocytes (at S2b) and Nebenkerns in spermatids in the onion stage were co-localized with F-actin (Figure 3(A1) and (B1), respectively). Thus, we examined whether the characteristic distribution of mitochondria depends on F-actin. Mitochondria accumulated next to the nucleus in S2b, as seen in the untreated cells (Figure 3(C1)). Some of the mitochondria had not accumulated in the cytoplasmic space next to the nucleus in the cells lacking F-actin (Figure 3(C1)). In S6, when the most elongated mitochondria were observed in the untreated cells, the appearance of granular mitochondria was noted after the treatment (Figure 3(D1)), indicating that depolymerization of F-actin inhibited the formation of the characteristic mitochondrial network structures. By contrast, the network structure, consisting of elongated mitochondria, was maintained in the meiotic cells lacking F-actin (as shown by the arrow in Figure 3(E2)), while the intracellular distribution of mitochondria was partially affected: some mitochondrial clusters were not accumulated on the equator below the cell cortex, unlike in the untreated cells. The drug-treated cells at late anaphase I (Figure 3(F1)) and telophase I (Figure 3(G1)) kept mitochondria on the central spindle microtubules, but not in a uniformly distributed fashion (30 cells/45 cells). Therefore, we conclude that F-actin is required for the subcellular localization of mitochondria during meiosis. After the drug treatment, abnormally shaped Nebenkerns, in which mitochondria were assembled homogenously, were observed (Figure 3(H1)) (111 cells/121 cells). F-actin is also required to produce the characteristic mitochondrial morphology and distribution before and during meiosis and for Nebenkern formation after meiosis.

### 2.3. Formation of the Elongated Mitochondria Was Inhibited by the Knockdown of the Fusion Factors and the Mitochondrial Morphology Proteins

We next investigated whether the mitochondrial fusion factors, Marf and Opa1are involved in the formation of elongated mitochondria in the spermatocytes of the cell growth phase. We also examined whether the knockdown of the mitochondrial morphology proteins Letm1 and EndoB, influences the formation of the structure. Ectopic expression of dsRNAs against *Opa1*, *Marf*, *Drp1*, and *EndoB* mRNAs in the spermatocytes decreased the relative quantity of mRNAs to less than 10% of that for the controls (Appendix A). In normal spermatocytes at the early stage (S2b) of the cell growth phase, mitochondria formed an elongated structure extending to 1 μm or longer and were clustered next to the nucleus (Figure 4(A1)). By contrast, in both *Marf*-depleted spermatocytes and *Opa1*-depleted cells, short and granule-shaped mitochondria (approx. 0.2 μm in length), dispersed throughout the cytoplasm without accumulating on the cells (all *n* > 160 *MarfRNAi^JF^*, *MarfRNAi^GD^*, and *Opa1RNAi^KK^*) (Figure 4(B2,C2)). Meanwhile, when a fission factor, Drp1, was downregulated via the knockdown of *Drp1* (*Drp1RNAi^JF^*) or ectopic expression of a dominant-negative mutant, *Drp1^DN^*, in spermatocytes, these cells maintained the clustering of mitochondria at S2b, as seen in the control. Regarding the shortened mitochondrial structure in S3 and S4 in the control, we were unable to detect any distinct differences in the mitochondrial morphology in the cells, wherein Drp1 was downregulated.

As the cell growth phase progresses, mitochondria formed an elongated network structure by the onset of meiosis in normal cells (Figure 4(F1)). In contrast, small granule-shaped mitochondria were observed in all cells after knockdown of the fusion factors *Marf* and *Opa1* (Figure 4(G1,H1)) (*n* > 325 cells in *MarfRNAi^GD^* and *Opa1RNAi^KK^*). In the S6 phase immediately before the onset of meiosis, normal cells formed elongated mitochondria with lengths exceeding 2 μm (Figure 4(F1)). In contrast, the largest spermatocytes subjected to *Marf*-knockdown and *Opa1*-knockdown contained mitochondria in granular form that were approximately 0.2 μm in length (Figure 4(G1,H1)). Furthermore, super-resolution microscopy observations confirmed the absence of the elongated mitochondria in the *Opa1RNAi^KK^* cells (Appendix A). Therefore, we conclude that the fusion factors Marf and Opa1 are required for the formation of the elongated mitochondria at S2b and S6. In contrast, the largest spermatocytes subjected to knockdown of the fission factor, Drp1 (*Drp1RNAi^JF^*), and those exhibiting ectopic expression of a dominant-negative mutant, *Drp1^DN^*, contained the elongated mitochondria seen in normal cells (see Appendix A(B1) for the mitochondria in the *Drp1^DN^*-expressing cells).

*EndoB*-depleted spermatocytes and the *Letm1*-depleted cells possessed shortened and granule-shaped mitochondria shorter than 1 μm at S2b and S6. These abnormally shaped mitochondria were also dispersed throughout the cytoplasm in the spermatocytes subjected to knockdown of these factors, even at the most developed stage (Figure 4(I2,J2)) (all of *EndoBRNAi^GD^* and *Letm1RNAi^GD^* cells (*n* > 523 cells examined in each)—see the super-resolution microscopic images in Appendix A(C1)). These mitochondrial morphology proteins are also required for the formation of elongated mitochondria.

### 2.4. Knockdown of the Fusion Factors and the Morphology Proteins Inhibited ATP Synthesis in the Spermatocytes

As the formation of elongated mitochondria, which retain increased ATP synthesis capabilities, was affected by the knockdown of fusion factors and morphology proteins, we examined whether ATP levels decreased in the testes in question (Appendix A). The average ATP amount in the control testes was 65.6 ± 8.4 mmol/mg. When *blw*, encoding the mitochondrial ATP synthase alpha-subunit, was depleted in the testis cells, the mean ATP amount decreased by 7.9 ± 0.3 mmol/mg. Thus, we next quantified ATP amounts in the testes subjected to knockdown of *Marf*, *Opa1*, *EndoB*, and *Letm1*. Those in the testes subjected to knockdown of the mitochondrial fusion factors decreased by up to 45% with respect to those of control, as follows: 30.9 ± 0.7 mmol/mg in *MarfRNAi^JF^*, 35.9 ± 0.5 mmol/mg in *MarfRNAi^GD^*, 29.2 ± 0.4 mmol/mg in *Opa1RNAi^HMS^*, and 31.2 ± 0.5 mmol/mg in *Opa1RNAi^KK^*. Consistently, the mean levels in testes subjected to knockdown of the morphology proteins also decreased as follows: 6.2 ± 0.3 mmol/mg in *EndoBRNAi^KK^*, 7.0 ± 0.3 mmol/mg in *EndoBRNAi^GD^*, 10.5 ± 0.2 mmol/mg in *Letm1RNAi^GD^*, and 10.4 ± 0.6 mmol/mg in *Letm1RNAi^HMS^*. These data suggest that ATP synthesis in mitochondria was reduced due to the knockdown of mitochondrial fusion factors and the morphology proteins.

### 2.5. Spermatocyte Growth Was Affected by the Knockdown of Mitochondrial Fusion Factors and the Morphology Proteins

The average diameter of the most-developed spermatocytes for which *blw* was knocked down was 27.0 ± 0.3 µm (as shown at the bottom of Figure 5). This is considerably smaller than that of the control spermatocytes at S6 (average 38.1 ± 0.3 µm) (the top of Figure 5). Thus, we investigated whether the reduced ATP levels in the spermatocytes for which mitochondrial fusion factors and the morphology proteins had been knocked down resulted in inhibition of the cell growth before meiosis. In contrast to the remarkable cell growth in normal spermatocytes, the maximum diameter was 30.3 ± 0.2 µm in spermatocytes from the *sa* mutants, wherein progression of cell growth is arrested in the middle of the growth phase. Similarly, it was 32.2 ± 0.3 µm in *twe* mutants, wherein the meiotic cell cycle is arrested just before the onset of meiosis. The average diameters of spermatocytes subjected to knockdown of the fusion factors at the most developed stage were as follows: 29.7 ± 0.4 µm for *MarfRNAi^JF^*, 29.6 ± 0.2 µm in *MarfRNAi^GD^*, 31.6 ± 0.3 µm in *Opa1RNAi^HMS^*, and 29.8 ± 0.4 µm in *Opa1RNAi^KK^*. The average diameters of these knockdown cells were as small as those of the *sa* mutant cells. Consistently, *Letm1*-depleted spermatocytes at the most developed stage (23.7 ± 0.2 µm for *Letm1*RNAi*^HMS^*, and 23.6 ± 0.7 µm for *Letm1RNAi^GD^*) were significantly smaller than normal spermatocytes at S6. Knockdown of the mitochondrial fusion factors Marf and Opa1 and the morphology proteins EndoB and Letm1 affected the cell growth of spermatocytes before male meiosis.

### 2.6. Spermatid Phenotype Arising from Abnormal Chromosome Segregation During Meiosis Owing to Knockdown of Fusion/Fission Factors and Morphogenetic Proteins

While the *blw* knockdown inhibited spermatocyte cell growth, the cells subjected to knockdown completed both meiotic divisions and formed normal-looking cysts consisting of 64 spermatids. Thus, we next investigated whether the knockdown of mitochondrial fusion and fission factors would affect the execution of meiosis. Sixty-four spermatocytes containing single nuclei of the same size were produced in normal testes (Figure 6(A1)). In contrast, 20% of intact cysts of spermatids derived from *Marf*-depleted spermatocytes (*MarfRNAi^GD^*) were composed of 33 to 63 cells (Figure 6(B1), Table 1). Consistently, all of the intact spermatid cysts from spermatocytes with another fusion factor, *Opa1* (*Opa1RNAi^HMS^*), knocked down, contained only 16 cells, each with a single nucleus that was larger than in controls and a Nebenkern. A total of 15% of the intact cysts were composed of 16 to 31 cells among the spermatid cysts that differentiated from *Opa1*RNAi*^KK^* spermatocytes (Figure 6(C1), Table 1). These phenotypes resembled those of the *twe* mutant males, wherein spermatocytes differentiated without one or both meiotic divisions [24]. The spermatids derived from the *Drp1*-depleted spermatocytes (*Drp1RNAi^JF^*) formed cysts made up of 64 normal-looking cells (all contained in the 59 cysts examined) (Table 1). In contrast, among spermatid cysts derived from spermatocytes expressing the dominant-negative Drp1 mutant Drp1^DN^, 60% of the intact spermatid cysts derived from spermatocytes expressing a dominant-negative mutant for *Drp1* (*Drp1^DN^*) consisted of 33–63 cells (30 cysts/51 cysts examined) (Figure 6(D1), Table 1). These results suggest that meiosis was not correctly completed in some of these spermatocyte cysts. Consistent with the results obtained following the knockdown of the fusion and fission factors, 90% of spermatid cysts from the *EndoB*-depleted spermatocytes and all spermatid cysts from the *Letm1*-depleted spermatocytes consisted of only 16 cells (Figure 6(E1,G1), Table 1), suggesting that both meiotic divisions did not take place in the *EndoB-* and *Letm1*-depleted spermatocytes. In summary, knocking down mitochondrial fusion and fission factors, along with morphology proteins, in spermatocytes impacted male meiosis.

Furthermore, in testes lacking mRNA for mitochondrial fusion factors, cysts containing abnormal spermatids resulting from failed chromosome segregation in meiosis were observed. The spermatids derived from the spermatocytes of *MarfRNAi^GD^*, *Opa1RNAi^HMS^*, and *Opa1RNAi^KK^* contained two or more nuclei, which were smaller than the nuclei of normal spermatids (note the arrowheads in Figure 6(C1) for an example). The nuclei of the *Opa1RNAi^HMS^* and *Opa1RNAi^KK^* cells were often different in size among spermatids within the same cysts (72.9% and 30.0%, respectively) (see the arrows and arrowheads in Figure 6(C2), Table 2). These phenotypes occur when chromosome segregation is arrested prematurely, resulting in the formation of small nuclei on lagging chromosomes.

### 2.7. Knockdown of the Fusion Factors and the Morphology Proteins Inhibited Cdk1 Activation Before the Onset of Meiosis

To determine why meiotic divisions failed in the spermatocytes subjected to knockdown of mitochondrial fusion factors, we investigated Cdk1 activation (Figure 7). Anti-MPM-2 immunostaining of normal spermatocytes at the mature stage (S6) showed intense signals in the nucleolus within the nucleus (as shown by the arrow in Figure 7(A1)), corresponding to the phosphorylation of nucleolar proteins by Cdk1. In contrast, among the largest spermatocytes subjected to knockdown of the mitochondrial fusion factors Marf and Opa1, a few cells exhibiting strong immunostaining signals on the nucleolus were observed (Figure 7(B1–E1)). In some *Opa1*-depleted spermatocytes, we observed multiple small dot-like signals within the nucleus (Figure 7(D2,E2)). However, these were not as prominent as the signals seen in the control cells. Consistently, in *EndoB*- and *Letm1*-depleted spermatocytes, MPM-2 punctate signals were observed in a less prominent form in locations that did not correspond to nucleoli (Figure 7(F1,G1)). However, we observed similar MPM2 signals in the control cells, even in the largest spermatocytes with *Drp1* knocked down (*Drp1RNAi^JF^*) or expressing the dominant-negative mutant, *Drp1^DN^* (>100 cells/>10 cysts examined). The above observations suggest that Cdk1 was not fully activated in the premeiotic cells, where the mitochondrial fusion and fission factors and EndoB and Letm1 were depleted, a finding consistent with the conclusions inferred from the meiotic phenotype that appeared in the *MarfRNAi* and *Opa1RNAi* cysts.

### 2.8. Inhibition of Meiotic Initiation Caused by Knockdown of the Fusion Factors and the Morphology Proteins Was Partially Rescued by Overexpression of Cyclin B

Next, we addressed a mechanism by which the knockdown of mitochondrial fusion factors resulted in the inhibition of Cdk1 activation. Cyclin B (CycB) was barely expressed in the middle stage (S4) in the growth phase of spermatocytes (Figure 8(A1)). Intense CycB signals were observed in both the cytoplasm and nucleus by the end of the growth phase (S6), which occurs before meiosis begins (Figure 8(B2)). In contrast, in the spermatocytes subjected to knockdown of *Opa1* and *Marf* (*Opa1RNAi^HMS^* and *MarfRNAi^GD^*), no spermatocytes with distinct CycB accumulation were found (>80 cells (>5 cysts) were examined) among the largest spermatocytes (Figure 8(C2,D2)). Similarly, in the spermatocytes wherein *EndoB* and *Letm1* were depleted, no distinct CycB accumulation was found (Figure 8(E2,F2)). However, we did not find significant changes in CycB accumulation in the cytoplasm and/or nucleus, even in the largest spermatocytes subjected to *Drp1* knockdown (*Drp1RNAi^JF^*) or expressing a dominant-negative mutant (*Drp1^DN^*) (>100 cells (>10 cysts) were examined). These results are consistent with the finding that no cells harboring active Cdk1 were observed in the testes, wherein mitochondrial fusion factors and morphology proteins were depleted.

To investigate whether the failure of meiotic initiation in the spermatocytes subjected to mitochondrial fusion factor knockdown was due to reduced CycB expression, we induced the ectopic overexpression of CycB in the cells and examined whether this action rescued the failure of meiosis (Table 3). Used as a negative control, the expression of *mCherry* led to the formation of 16-cell spermatid cysts at a high rate (84.3% among 108 cysts), but no 64-cell cysts. were observed in the testes with *Opa1* knocked down. In contrast, when *CycB* expression was simultaneously induced, the frequency of 16-cell cysts decreased (51.9% among 129 cysts) and, instead, 64-cell cysts were observed (18.0% among 129 cysts). Consistently, spermatocytes with *EndoB* knocked down and exhibiting *mCherry* expression formed 16-cell spermatid cysts at a high frequency (62.6% among 107 cysts examined), but there were no 64-cell cysts. In contrast, when *CycB* expression was simultaneously induced, the frequency of 16-cell cysts decreased (7.6% among 107 cysts), and, instead, 64-cell cysts appeared (27.6% among 105 cysts). Therefore, we conclude that the failure of meiosis caused by the knockdown of *Opa1* or *EndoB* was partially restored by the CycB overexpression.

### 2.9. Knockdown of the Fusion and Fission Factors and the Morphology Proteins Caused Abnormalities in Nebenkern Formation

In a normal spermatid, mitochondria assemble into a single spherical structure called a Nebenkern after meiosis is completed (Figure 9(A1)). In contrast, we found incomplete Nebenkerns that failed to form a sphere in the spermatids that developed from the spermatocytes subjected to knockdown of *Marf* (*MarfRNAi^GD^*) (Figure 9(B1)), and *Opa1* (*Opa1RNAi^HMS^*, *Opa1RNAi^KK^*) (Figure 9(C1,D1)). Most of the Nebenkerns exhibited uneven staining, with areas in the center remaining unstained or less stained (Figure 9(B1–E1)). (all of the *MarfRNAi^GD^* spermatids (*n* > 325 examined), and more than 70% of those corresponding to *Opa1RNAi^HMS^* and *Opa1RNAi^KK^* (*n* > 952 cells). Despite being less frequent, a similar phenotype in Nebenkern morphology was also observed in spermatids expressing *Drp1^DN^* (Figure 9(E1)). Among the spermatids in which *EndoB* or *Letm1* was depleted, more severe Nebenkern phenotypes, such as multiple smaller Nebenkerns per spermatid (as shown by the arrows in Figure 9(C1,F1,G1)) and unassembled mitochondria that were not integrated into a Nebenkern in the cytoplasm (Figure 6(H2)), were frequently observed in the spermatids. These observations suggest that Nebenkern formation was affected by the knockdown of mitochondrial fusion and fission factors and EndoB and Letm1 (Figure 9(F1–I1)).

## 3. Discussion

### 3.1. Mitochondria in Drosophila Spermatocytes Undergo Stage-Specific Changes Between a Shortened Form and an Interconnected Network Structure

Several studies have reported that mitochondria undergo a remarkable morphological change to form a large cluster called a Nebenkern after meiosis [18,20,21]. We focused on investigating mitochondrial fine structures in spermatogonia and spermatocytes before and during meiosis using high-resolution confocal microscopy, super-resolution microscopy, N-SIM, and TEM. Spermatogonia derived from germline stem cells undergo four mitotic divisions, during which shortened and granular mitochondria can be observed. This morphology is consistent with that observed in mammalian cells during mitosis [7,25]. However, we have demonstrated that a mitochondrial network structure composed of elongated mitochondria alternates with shortened mitochondria, as spermatocytes develop. In the S2b stage corresponding to polar spermatocytes, mitochondria accumulate next to the nucleus [26]. Illustrations based on previous EM observations show discrete mitochondria with a granular shape or oval capsules that look like two fused entities (Tate’s TEM observation in [18]). Whereas previous studies conducted using light microscopy also noted that phase-dark structures possibly corresponding to mitochondria form clusters at this stage [9,22], we have performed immunostaining experiments to identify mitochondria in meiotic cells [19,20]. Our observations are consistent with previous descriptions. In Tate’s illustration based on his EM observations, an elongated mitochondrion-like structure is depicted in the most developed spermatocytes [18]. Moreover, in combination with the previous immunostaining results, we conclude that the organelles are associated with microtubule and nuclear envelope structures that form specifically during meiosis. Mitochondria are subsequently released into the cytoplasm upon cytokinesis. This specific pattern of mitochondrial distribution suggests that a regulatory mechanism ensuring equal segregation of mitochondria transferred to daughter cells during meiosis, rather than an equal partitioning of the cytoplasm, may exist.

In mammalian cell mitosis, the mitochondrial membrane potential changes during cell division. Mitochondria are polarized until metaphase to the same extent as in interphase cells, but, thereafter, they become depolarized. After cytokinesis, the potential is recovered. The supply of electrons to the mitochondrial electron transfer chain is suppressed during M phase, possibly to reduce the quantity of reactive oxygen species (ROS) generated by mitochondrial fragmentation during M phase [7,27]. In contrast, during meiosis in *Drosophila* males, mitochondria maintain the network structure formed during the cell growth phase. Since active mitochondria are transferred to daughter cells while staying intact, ATP production continues uninterrupted. This may give an advantage in meiosis, as ATP can be provided immediately on-site. ATP is also required for morphological changes in mitochondria after the completion of meiosis II [22].

### 3.2. Requirement of Fusion Factors, Microtubules, and F-Actin for the Formation of Elongated Mitochondrial Networks Constructed Before and During Male Meiosis

This study demonstrates that Marf, which is required for the fusion of the mitochondrial outer membrane, and Opa1, which is necessary for inner-membrane fusion, are indispensable for the formation of elongated mitochondrial structures at S2b and S6. This finding is consistent with observations made in *Drosophila* axons and heart tubes [28,29]. When the fusion/fission balance shifts toward fission because of a decrease in fusion, the network converts a higher number of smaller mitochondria in neurons [29,30], because Drp1 counteracts Marf and Opa1 [28]. When the fission factor, Drp1, was depleted using *Drp1RNAi* stock that enables effective knockdown, or when Drp1^DN^ was ectopically expressed, no cells with remaining elongated mitochondrial structures were observed in the subsequent stages, wherein the elongated structure is shortened. A previous study reported that the phase-dark structures corresponding to mitochondria form a tight cluster in the spermatocytes homozygous for the lethal allele of Drp1 [22]. The cells with *Drp1RNAi* and expressing *Drp1^DN^* may have retained more Drp1 activity relative to spermatocytes homozygous for the lethal mutation. These mitochondrial network structures are typical of normal cells before meiosis, so even if the cluster increases in cells with lower Drp1 activity, they may be difficult to find.

In addition to the well-known GTPases serving as mitochondrial dynamics factors, knockdown of EndoB, which is localized to the outer membrane (OMM), and Letm1, which is localized to the inner membrane, affected the formation of the mitochondrial network. Super-resolution microscopy observations revealed that mitochondrial phenotypes seem qualitatively different when comparing the knockdown of the future factors and the morphology proteins required for membrane morphology. In mammalian cultured cells, inhibiting EndoB1 results in an abnormal structure where only the OMM elongates [13]. Similarly, the mitochondrial morphology and cristae structures are disturbed in Letm1RNAi mammalian cells [12]. These two mitochondrial morphology proteins are also required for the formation of the mitochondrial network before meiosis in *Drosophila*.

Contrary to our initial predictions, we did not observe clear morphological changes in mitochondrial elongation or network formation after microtubule depolymerization. Microtubules are crucial for mitochondrial dynamics, specifically in mitochondrial fission and fusion in mammalian cells [30,31,32,33]. Similarly, the formation of mitochondrial clusters requires Milton-dependent mitochondrial transport in *Drosophila* polar spermatocytes, although there is no direct evidence that microtubules control the fusion or formation of elongated mitochondrial networks [34]. Since Milton is a myosin-binding protein with microtubule motor activity, it is reasonable to speculate that microtubules influence mitochondrial dynamics. However, we cannot rule out the notion that there was not enough time for detectable morphological changes in mitochondria to occur. Given the existence of a conflicting report indicating that fusion of mitochondria in mammalian cells is independent of microtubules [35], it cannot be ruled out that the involvement of microtubules may differ depending on the type of cell.

F-actin colocalized with mitochondrial clusters in the polar spermatocytes and with Nebenkerns in spermatids. When F-actin polymerization was inhibited, more pronounced abnormalities appeared in the mitochondrial structures relative to those treated with microtubule inhibitors. Notably, in the mature spermatocytes treated with F-actin inhibitors, mitochondrial network structures were not observed. In meiotic cells, although mitochondrial distribution was partially affected, the network structures themselves remained intact. Once constructed, the structure may remain relatively stable during meiosis. While no studies have directly investigated the role of F-actin in *Drosophila* early spermatogenesis, a recent study using mammalian cells has shown that mitochondria-associated F-actin is necessary for both mitochondrial fusion and fission to mark the future sites where mitochondrial rearrangements will occur [36]. This result is consistent with our current findings. F-actin may play a more critical role in maintaining and/or constructing the structure of the mitochondrial network.

### 3.3. Requirement of the Mitochondrial Network to Be Formed via Fusion Factors for the Cell Growth of Spermatocytes Before Meiosis

Knockdown of mitochondria fusion factors and the mitochondrial morphology proteins, EndoB and Letm1, impaired ATP synthesis in the mitochondria. In these spermatocytes, cell growth was inhibited. Similar growth inhibition was observed upon knockdown of blw, which encodes an ATP synthase. Therefore, ATP depletion caused by inhibiting fusion factors and morphology proteins may result in growth inhibition. ATP is synthesized more efficiently in elongated mitochondria formed via these factors [37]. It is reasonable to speculate that inhibition of this process led to a decrease in ATP levels and cell growth inhibition. In ATP-depleted cells, AMPK activity increases, and inhibiting Tsc2 in the insulin-like peptides (ILP)-induced pathway eventually reduces protein synthesis [38]. Activation of the signaling pathways by ILPs is required for cell growth before meiosis [39]. Consistently, when mitochondrial dynamics are disturbed in yeast and mammalian cells, ATP synthesis ceases [40,41]. Therefore, inhibition of elongated mitochondrial structures, formed via mitochondrial fusion factors and morphology proteins, may have led to a decrease in ATP levels, resulting in cell growth impairments in spermatocytes. In normal cells, elongated mitochondria were observed in the polar spermatocyte stage (S2b). During the subsequent apolar stage (S3–S5), the spermatocyte growth becomes most active. Before these stages, ATP required for these processes may be efficiently synthesized within the elongated network structure. 

### 3.4. Elongated Mitochondrial Networks Are Transferred to Daughter Cells While Maintaining the Structures During Male Meiosis

In the interphase in mammalian cultured cells, mitochondria form an interconnected tubular network, and their membrane potential increases from G1 to S phase and during G2/M phase [37]. Subsequently, mitochondria undergo fission before the M phase. This process is assumed to facilitate the equal partition of mitochondria [2]. In contrast, we demonstrated that the structure of the mitochondrial network is established in spermatocytes and is transferred while maintaining its structure in meiosis. Compared to the shortened forms (where ATP synthesis is suppressed), the fused forms allow ATP to be supplied when it is needed. This is beneficial during the ATP-dependent dynamic stages of the cell growth phase and subsequent M phase. Additionally, this interconnected structure may facilitate equal distribution of the organelles during meiosis. Afterward, the organelles quickly cluster together to create a Nebenkern. In mitosis, even if one of the two daughter cells receives an insufficient number of mitochondria, the mitochondria can be amplified during the next S phase. In contrast, meiosis lacks a subsequent cell cycle with which to compensate for the insufficiency. To prevent this problem from occurring, there may be a control mechanism that guarantees equal mitochondrial partitioning.

Mitochondria are transferred into daughter cells in male meiosis, while being closely associated with aster and central spindle microtubules (live analysis in this study [18,19,26]). Unexpectedly, despite the absence of microtubules, the mitochondria remained accumulated next to the nucleus in the polar spermatocyte, and remained distributed in the cytoplasm, maintaining the clusters, although they lost the orientation toward the spindle poles. Once the mitochondrial network is formed, the cluster may be maintained without disintegrating even in the absence of microtubules. This notion may be inconsistent with reports on mitosis in mammalian cells [30]. Both cytoskeletons play indispensable roles in transporting mitochondria to specific sites in mammalian mitotic cells and neurons [7,42,43]. By contrast, when F-actin polymerization was inhibited, some mitochondrial clusters left their original locations in polar spermatocytes and the meiotic cells. F-actin may play a more significant role in distributing and moving along microtubules and clustering at the cell equator during meiosis. Each mitochondrion in the network may be closely interconnected with others. Alternatively, other intracellular structures, such as F-actin, may bundle mitochondria.

### 3.5. A Possible Checkpoint Mechanism That Monitors Mitochondrial Morphology and/or Function and Prevents the Progression of Meiosis in the Presence of Abnormalities

This study demonstrates that the network structure composed of elongated mitochondria is established before meiosis in *Drosophila* males and is transferred to daughter cells, maintaining their network structures. This distribution differs from that in mitosis. After depleting fusion factors and the morphology proteins, EndoB and Letm1, we observed abnormal spermatid cysts consisting of only 16 cells. This phenotype has been observed in other mutants, in which the first and second meiosis do not occur [23,44,45]. The inhibition of fusion factors results in a shift in the fission/fusion balance toward fission, and the resultant inability to form a network structure leads to fewer meiotic cells. However, in testes where the fission factor Drp1 was downregulated—but not in testes with Drp1 knocked down—abnormal spermatid cysts derived from a single meiosis were observed, albeit at a lower frequency. Therefore, the fission factor may also influence male meiosis through maintaining the fusion/fission balance.

Moreover, these abnormal cells in spermatid cysts, consisting of 16 or 32 cells, possessed two or more nuclei that were smaller than nuclei containing diploid chromosome complements. In the absence of meiosis I initiation, chromosome segregation does not happen [23], and multiple nuclei should not form. This phenotype bears a striking resemblance to a mutant phenotype in which chromosome separation commences yet is abruptly halted during anaphase, resulting in the formation of nuclei at the sites of the chromosomes [44]. When the elongated structure is disrupted, meiosis initiates but may fail to terminate at the appropriate time. In the spermatocytes subjected to knockdown of factors involved in mitochondrial dynamics, CycB accumulation at the onset was less pronounced than in controls. No nuclear transport of CycB nor, consistently, prominent Cdk1 activation was observed. In the cells subjected to knockdown, the onset of meiosis may have been delayed relative to the controls. In mouse oocytes, Drp1 knockdown also affects the restart of meiosis [46]. The timing of CycB expression in spermatocytes is determined by transcriptional control and the release from mRNA translation suppression. In future studies, it will be important to investigate whether the Rbp4 and Fest proteins [47], which regulate CycB translation, are affected by the knockdown of the fusion factors. In ATP-synthetase-knockdown spermatocytes, meiosis is completed ([48], this study). Therefore, impaired ATP synthesis in mitochondria is not a direct cause of failure and abnormalities in meiosis. This study does not clearly explain why the knockdown of the factors involved in mitochondrial morphological changes impacts chromosome segregation during meiosis. This is a limitation of this research.

### 3.6. Conclusions

This study confirmed that the mitochondrial fusion factor Opa1 is essential for forming the structure of the mitochondrial network in early *Drosophila* spermatogenesis and revealed that another fusion factor, Marf, and the mitochondrial morphology proteins LetM1 and EndoB are also required. Inhibition of network formation by knocking down of these factors resulted in reduced ATP synthesis in spermatocytes, inhibited cell growth, and the prevention of meiosis progression. Cdk1 was not activated, presumably because CycB was not exported from the nucleus before the onset of meiosis. This finding suggests that if mitochondrial status does not meet certain conditions before energy-demanding cell growth and meiosis, meiotic progression may be prevented. Our findings, suggesting the existence of a meiotic checkpoint, can provide important insights into spermatogenesis research.

## 4. Materials and Methods

### 4.1. Drosophila Stocks

To silence the mRNAs of mitochondrial dynamic factors and morphology proteins, we used the *UAS-RNAi* stocks listed in Appendix A. To induce ectopic expression of dsRNAs against the relevant mRNAs, we used *P{UAS-Dcr2}*; *P{bam-GAL4::VP16}* (hereinafter abbreviated as *bam-GAL4*) for spermatocyte-specific *RNAi* experiments [49]. Thus, spermatocytes carrying *UAS-XRNAi^#^*, *P{UAS-Dcr2}*, and *P{bam-GAL4::VP16}*, in which the mRNA for the *X* gene is depleted, are denoted as *XRNAi^#^*. As a control, F1 males between *bam-Gal4* and wild-type (*bam>+*) were used. For the downregulation of Drp1, the activity of its dominant-negative mutant protein was also induced using *P{UAS-Drp1^K38A^}* (*UAS-Drp1^DN^*) [50]. We used *M{UAS-CycB.ORF.3xHA}* for the overexpression of Cyclin B [51]. To visualize microtubules, *P*{*Ubi-tub56D-GFP*} (*GFP-tubulin*) was used [52]. *P{bam-GAL4::VP16}* was used as a Gal4 driver for spermatocyte-specific induction of gene expression [53]. All *Drosophila* stocks were maintained on standard cornmeal food at 25 °C, as previously described [54]. To maintain the stocks and obtain adults for aging-related experiments, standard cornmeal fly food was prepared as follows: per liter of water, 40 g of dried yeast (Asahi Group, Tokyo, Japan), 40 g of corn flour (Nippun, Tokyo, Japan), 100 g of glucose (Kato Chemical, Aichi, Japan), and 7.2 g of agar powder (Matsuki Agar, Nagano, Japan) were added, and 5 mL of a 10% methyl para hydroxybenzoate solution and 5 mL of propionic acid (Tokyo Kasei Kogyo, Tokyo, Japan) were added to 1 L of the fly food. To efficiently induce GAL4-dependent gene expression, individuals carrying the GAL4 driver gene and UAS transgenes were raised at 28 °C. Other experiments and stock maintenance procedures were conducted at 25 °C.

### 4.2. Preparation of Post-Meiotic Spermatid Cysts

To estimate whether two consecutive meiotic divisions were executed correctly, we observed nuclei in post-meiotic spermatids at the onion stage just after the completion of meiosis II under phase-contrast microscopy, as previously described [51,55]. A pair of testes from pharate adults or newly eclosed adult flies (within 1 day old) was dissected to isolate spermatocyte cysts in Testis buffer (183 mM KCl, 47 mM NaCl, 10 mM EDTA, pH 6.8) and covered with a coverslip (Matsunami Co., Osaka, Japan) to flatten the cysts. To observe spermatids under a phase-contrast microscope, the cysts collected from the testes were mildly flattened in Testis buffer under a cover slip. After removing the coverslips, we transferred them into 100% methanol for 3 min at −30 °C to fix the samples. Subsequently, they were rehydrated in PBS (137.0 mM NaCl, 2.7 mM KCl, 10.1 mM Na_2_HPO_4_·12H_2_O, and 1.8 mM KH_2_PO_4_), and then the DNA was stained with DAPI. Samples were observed using a phase-contrast microscope (Olympus, Tokyo, Japan, model: IX81). Normal spermatocytes undergo two meiotic divisions to form 64 spermatocytes simultaneously. The resultant spermatid possesses a one-to-one ratio of nucleus to mitochondrial aggregates, known as Nebenkerns [18,55].

### 4.3. Administration of a Drug to the Testis Cells

To inhibit microtubule polymerization, testes from young adult flies were dissected in 50 µM/mL colchicine (# W01W0103-0385 (WAKO Pure Chemicals, Osaka, Japan)) in testis buffer and incubated for 15 min before fixation. For the inhibition of actin-filament polymerization, testes from adult flies were dissected in 100 µM of latrunculin A in testis buffer for 45 min before fixation. Testes squashes, conducted to evaluate onion-stage spermatids, were performed as described above and viewed under phase-contrast microscopy.

### 4.4. Immunostaining of Testis Cells

Testis cells, collected from the testes as described above, were fixed in ethanol at −30 °C for 10 min and then in 3.7% formaldehyde for 7 min. The slides were permeabilized in PBST (PBS containing 0.01% Triton-X) for 10 min and blocked with 10% normal goat serum in PBS. The following primary antibodies were used at the dilutions described: MPM-2 antibody (05-368, Sigma-Aldrich, St. Louis, MO, USA), 1/200; anti-Complex V alpha-subunit monoclonal antibody (#439800, Thermo Fisher, Waltham, MA, USA), 1/400; and anti-Cyclin B monoclonal antibody (Developmental Studies Hybridoma Bank, Iowa City, IA, 1/200. After incubating with the primary antibody overnight at 4 °C, the fixed samples were repeatedly washed in PBS and subsequently incubated with anti-mouse or anti-rabbit IgG conjugated with Alexa Fluor 488 or 555 (Thermo Fisher, Waltham, MA, USA). After being washed in PBS, they were mounted with VECTASHIELD Mounting Medium with DAPI (Vector Laboratories, Burlingame, CA, USA) and observed under a IX81 fluorescent microscope (Olympus, Tokyo, Japan). Image acquisition was controlled using the MetaMorph software version 7.6 (Molecular Devices, SAN Jose, CA, USA).

### 4.5. ATP Assay

Ten pairs of testes were homogenized in cell lysis buffer (10 mM Tris (pH 7.5), 100 mM NaCl, 1 mM EDTA, 0.01% Triton X-100) on ice. These homogenates were immediately frozen in liquid nitrogen and subsequently inactivated at 99 °C for 3 min. After centrifugation at 6010× *g* for 10 min, the ATP levels were quantified in the supernatants using the ATP Determination Kit (#A22066, Invitrogen, Waltham, MA, USA). Fluorescence intensity was measured using a luminometer (Lumat LB9507, Berthold Technologies, Bad Wildbad, Germany). Based on the standard curve created, the ATP levels of the samples were determined. Protein concentration was measured using the Qubit 2.0 Fluorometer (Invitrogen, Waltham, MA, USA) and calculated as ATP [mM]/Protein [mg].

### 4.6. Transmission Electron Microscope Observation of Adult Testes

Abdomens of young male flies (within two days after eclosion) were placed in fixative solution (4% paraformaldehyde, 2% glutaraldehyde in 0.1 M cacodylate buffer, pH = 7.4), as previously reported [56]. The fixed specimens were washed with 0.1 M cacodylate buffer and treated with 2% osmium tetroxide in 0.1 M cacodylate buffer. The specimens were dehydrated via consecutive incubation in 50%, 70%, 90%, and 100% ethanol. After the specimens were infiltrated with propylene oxide and put into a 7:3 mixture of propylene oxide and resin (Quetol-812, Nisshin EM Co., Tokyo, Japan), they were transferred to new resin and polymerized. The specimens in the polymerized resins were ultrathin-sectioned at 70 nm using Ultracut-UCT (Leica, Vienna, Austria) and mounted on copper grids. After being stained with 2% uranyl acetate, the sections were washed with distilled water and stained with lead stain solution (Sigma-Aldrich Co., Tokyo, Japan). The grids were observed using a transmission electron microscope (JEM-1400Plus, JEOL Ltd., Tokyo, Japan) at 100 kV acceleration voltage and photographed with a CCD camera (EM-14830RUBY2, JEOL Ltd., Tokyo, Japan).

### 4.7. Statistical Analysis

Each dataset comparing the control and each knockdown was statistically assessed using Student’s *t*-test, as described in a previous study [57]. Initially, an F-test was performed to determine equal or unequal variances, and Student’s *t*-test was employed to perform when the value was greater than 0.05 (equal variance). Data were considered significant at *p*-values < 0.05. Statistical analyses were performed using Excel (version 16.78.3, Redmond, WA, USA) and GraphPad Prism 9 (GraphPad Software, San Diego, CA, USA).

## Figures and Tables

**Figure 1 ijms-26-09991-f001:**
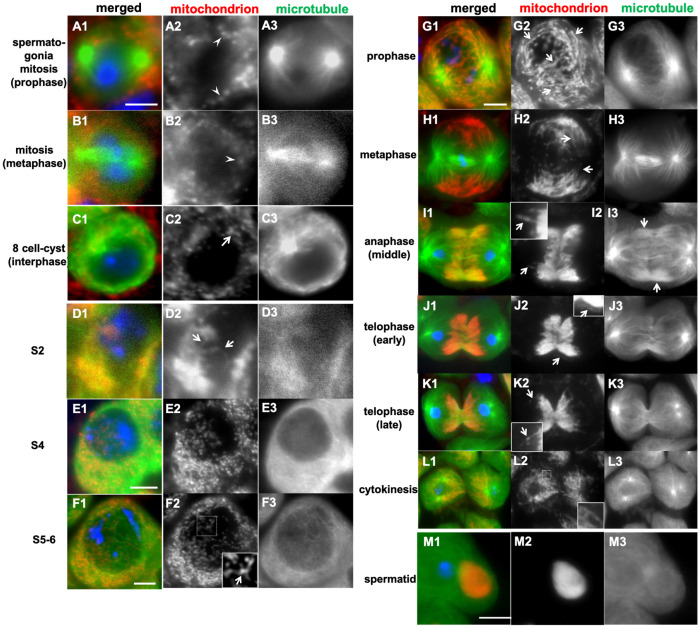
Differences in mitochondrial morphology between spermatogonia undergoing mitosis and spermatocytes before and during the first meiotic division in *Drosophila*. (**A1**–**M1**) Immunostaining of spermatogonia and spermatocytes expressing GFP-Tubulin (green in (**A1**–**M1**) and white in (**A3**–**M3**)) using Complex V alpha-subunit antibody (red in (**A1**–**M1**) and white in (**A2**–**M2**)). Blue in (**A1**–**M1**) indicates DNA. (**A1**–**C1**) Spermatogonia undergoing mitosis (**A1**,**B1**) and in interphase (**C1**). Mitochondria exhibit a granular morphology (arrowheads in (**A2**,**B2**)) once mitosis starts, or slightly larger clusters (arrow in (**C2**)) during interphase. (**D1**–**F1**) Spermatocytes during the cell growth phase. (**D1**) A spermatocyte at the S2b stage, in which elongated mitochondria are formed (arrows in (**D2**,**F2**)). (**E1**) A spermatocyte at the S4 stage. (**F1**) A spermatocyte at the S5–6 stages (inset: a magnified view of the region enclosed by a square in (**F2**)). (**G1**–**K1**) The primary spermatocytes undergoing meiotic division. A prophase I cell (**G1**), a metaphase I cell (**H1**), and anaphase I cells (**I1**,**J1**). Arrows in (**G2**–**J2**): elongated mitochondrial network structures. From telophase I (**K1**) to cytokinesis (**L1**). (**M1**) Upon completion of the second meiotic division, mitochondria form a Nebenkern. Insets in (**I2**,**K2**,**L2**): magnified views. Bars: 10 µm.

**Figure 2 ijms-26-09991-f002:**
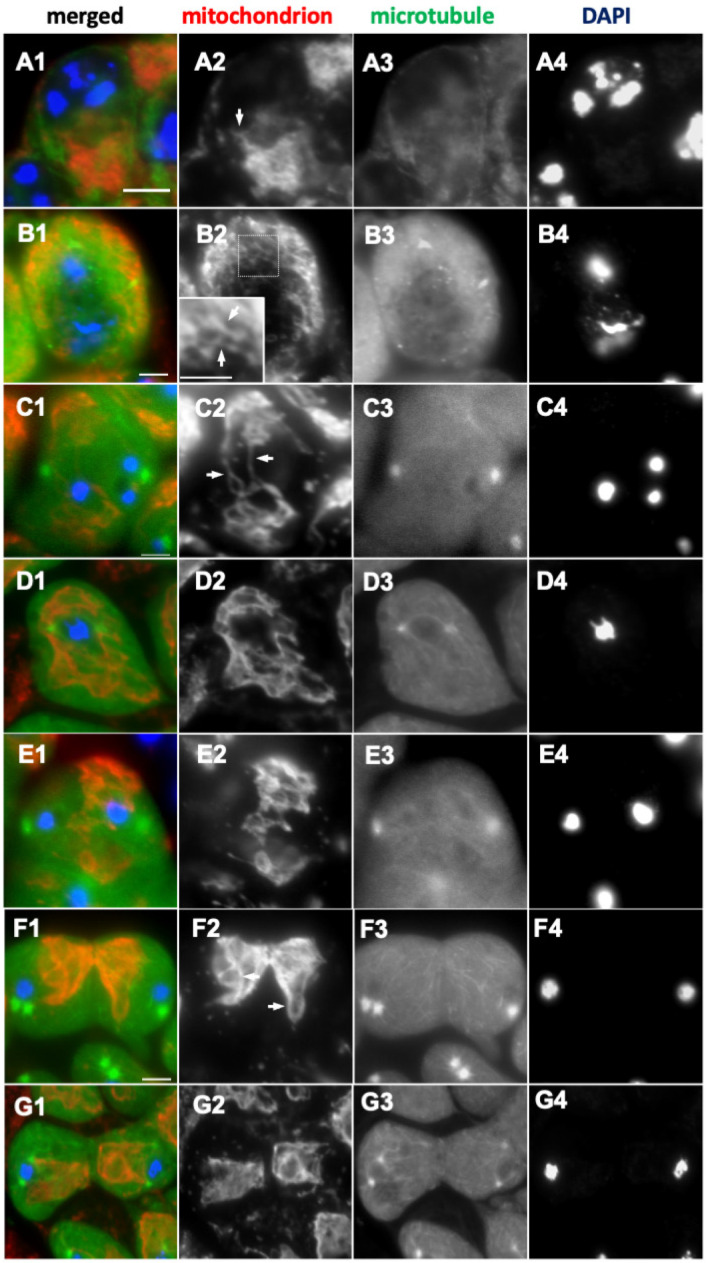
The distribution of mitochondria associated with microtubules in primary spermatocytes was disrupted by colchicine treatment. (**A1**–**G1**) Immunostaining of colchicine-treated primary spermatocytes with anti-Complex V alpha-subunit antibody conducted to visualize mitochondria (red in (**A1**–**G1**) and white in (**A2**–**G2**)). The cells expressing GFP-Tubulin (green in (**A1**–**G1**) and white in (**A3**–**G3**)) were used to confirm microtubule depolymerization. DNA (blue in (**A1**–**G1**) and white in (**A4**–**G4**)). (**A1**) A colchicine-treated spermatocyte at S2b. Arrow: The elongated mitochondria. (**B1**) A colchicine-treated spermatocyte in the S6 stage. (inset: enlarged view of the area enclosed by a square). (**C1**–**G1**) Mitochondria in colchicine-treated spermatocytes undergoing meiosis I in prophase-like (**C1**) and metaphase-like (**D1**) stages. Arrows in (**C2**): Mitochondria that remained localized in the cytoplasm while maintaining their network structure. (**E1,F1**) Cells that have passed through metaphase I/anaphase I before colchicine treatment. Arrows in (**F2**): Mitochondria whose orientations toward the microtubule-organizing center are lost. (**G1**) Telophase I cells. Bars: 10 μm.

**Figure 3 ijms-26-09991-f003:**
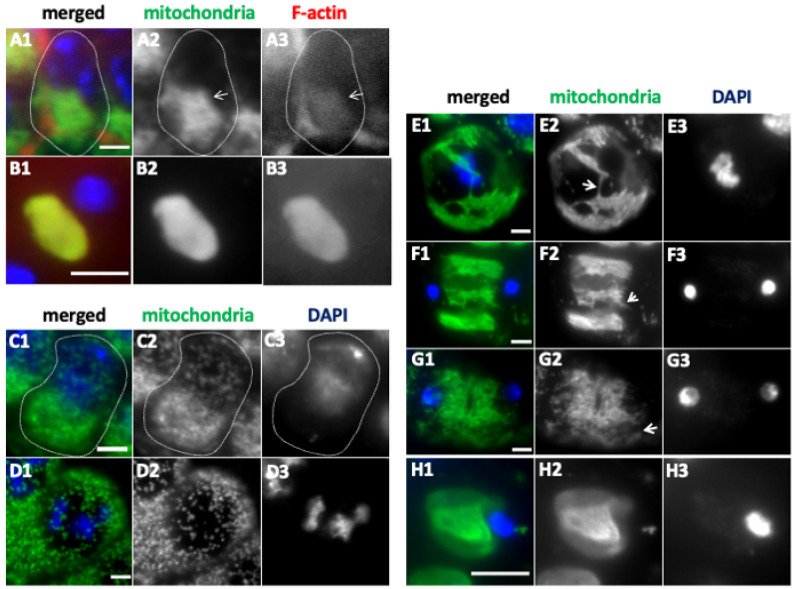
The close association between mitochondria and F-actin in spermatocytes in S2b of the growth phase and in early spermatids, and the perturbation of the distribution after Latrunculin A treatment. (**A1**,**B1**) Anti-Complex V alpha-subunit immunostaining conducted to observe the colocalization of mitochondria (green in (**A1**,**B1**) and white in (**A2**,**B2**)) with F-actin, visualized via phalloidin staining (red in (**A1**,**B1**) and white in (**A3**,**B3**)) of early spermatocytes at S2b (**A1**) and spermatids in the onion stage (**B1**). Arrow: elongated mitochondria in the S2b cell. A mitochondrial aggregate (Nebenkern) colocalizes with F-actin in the spermatid (**B1**). (**C1**–**H1**) Mitochondria in the cells treated with Latrunculin A were visualized via immunostaining (green in (**C1**–**H1**) and white in (**C2**–**H2**)). DNA staining by DAPI (blue in (**C1**–**H1**) and white in (**C3**–**H3**)). (**C1**) Mitochondria in a spermatocyte in S2b. (**D1**) Shortened granular-like mitochondria in the spermatocyte in S6. (**E1**–**H1**) Mitochondria in Latrunculin A-treated meiotic cells in metaphase I (**E1**), anaphase I (**F1**), and telophase I (**G1**), and a Nebenkern exhibiting heterogeneous staining in spermatids (**H1**). Bars: 10 µm.

**Figure 4 ijms-26-09991-f004:**
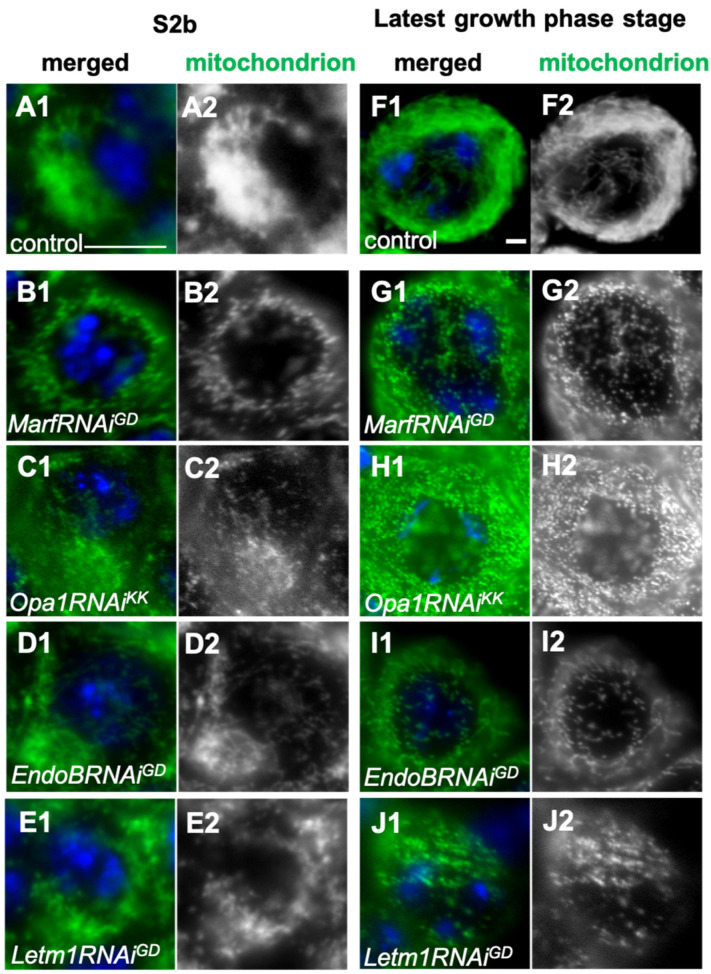
Knockdown of the mitochondrial fusion factor genes *Marf* and *Opa1* and that of *EndoB* and *Letm1* and their distribution in spermatocytes during the cell growth phase. (**A1**–**J1**) Immunostaining of early spermatocytes at S2b (**A1**–**E1**), and at S6 or the most developed stages of the growth phase (**F1**–**J1**) conducted to visualize mitochondria (green in (**A1**–**J1**) and white in (**A2**–**J2**)). Blue in A1–J1: DNA. (**A1**,**F1**) Wild-type (control) spermatocytes at S2b (**A1**) and S6 (**F1**). Spermatocytes at S2b (**A1**–**E1**) and S6 (**F1**–**J1**) with knockdown of the mitochondrial fusion factors Marf and Opa1 and the morphology proteins EndoB and Letm1. (**B1**,**G1**) *MarfRNAi^GD^*, (**C1**,**H1**) *Opa1RNAi Opa1RNAi^KK^*. Spermatocytes at S2b (**D1**,**E1**) and S6 (**I1**,**J1**) with knockdown of *EndoB* and *Letm1*. (**D1**,**I1**) *EndoBRNAi^GD^*. (**E1**,**J1**) *Letm1RNAi^GD^*. Bars: 10 μm.

**Figure 5 ijms-26-09991-f005:**
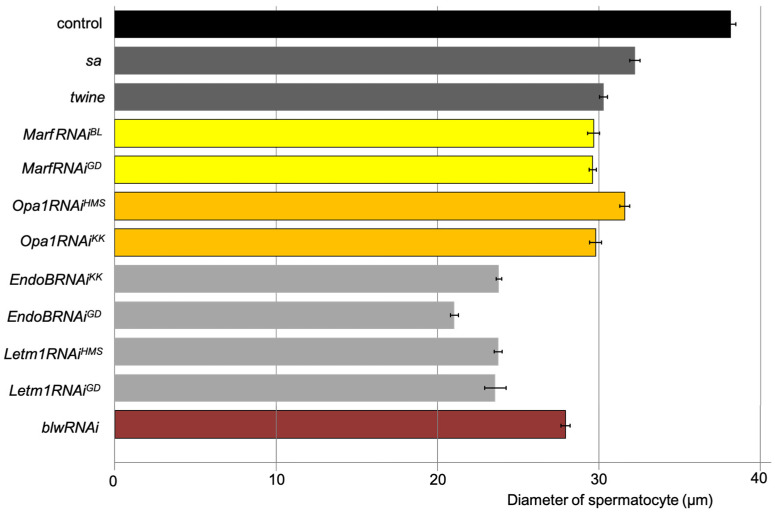
Growth inhibition of spermatocytes subjected to knockdown of mitochondrial fusion factors, the morphology proteins, and the *blw* gene encoding the mitochondrial complex V α subunit. The horizontal axis shows the mean diameter (µm) of the control spermatocytes at S6; the mean diameter of the cells at the most developed stages in which the mitochondrial fusion factor genes *Marf* (yellow: *MarfRNAi^JF^* and *MarfRNAi^GD^*) and *Opa1* (orange: *Opa1RNAi^HMS^* and *Opa1RNAi^KK^*) were knocked down; the cells in which the mitochondrial morphology genes *EndoB* (gray: *EndoBRNAi^KK^* and *EndoBRNAi^GD^*) and *Letm1* (gray: *Letm1RNAi^HMS^* and *Letm1RNAi^GD^*) were knocked down; and the cells in which *blw* was knocked down (brown: *blwRNAi*). The diameters of spermatocytes within the cysts at the S6 or the most developed stages (*n* ≥ 20 cells) were measured, and the average length is shown on the horizontal axis. *p* < 0.0001 in every genotype compared with the control (Student’s *t*-test). Error bars indicate the standard error of the mean.

**Figure 6 ijms-26-09991-f006:**
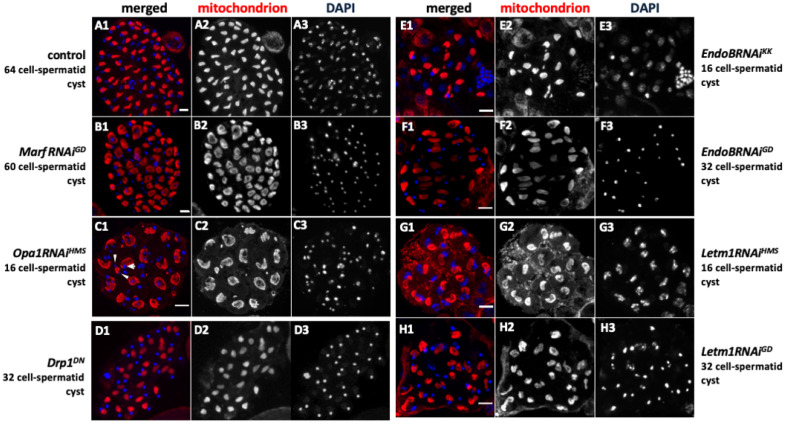
Abnormal spermatid cysts derived from the spermatocytes in which mitochondrial fusion and fission factors and EndoB and Letm1 were knocked down without undergoing either one of or both meiotic divisions. (**A1**–**H1**) Immunostaining was conducted to visualize mitochondria (red in (**A1**–**H1**) and white in (**A2**–**H2**)) in the spermatids of intact cysts at the onion stage. DNA staining (blue in (**A1**–**H1**) and white in (**A3**–**H3**)). (**A1**) An intact cyst composed of 64 spermatids in the control testes. (**B1**–**D1**) Intact spermatid cysts derived from spermatocytes in which *Marf* (**B1**; *MarfRNAi^GD^*) and *Opa1* (**C1**; *Opa1RNAi^GD^*, **D1**; *Opa1RNAi^HMS^*) were knocked down. (**B1**) An intact spermatid cyst composed of 60 cells in *MarfRNAi^GD^* testes. (**C1**) An intact cyst composed of only 16 cells in an *Opa1RNAi^HMS^* testis. The arrow in C1 indicates an example of normal-sized nuclei, and arrowheads indicate smaller nuclei. (**D1**) An intact cyst composed of only 32 cells in the testes expressing a dominant-negative mutant *Drp1^DN^*. (**E1**–**H1**) Intact spermatid cysts derived from spermatocytes subjected to knockdown of *EndoB* (**E1**; *EndoBRNAi^KK^*, **F1**; *EndoBRNAi^GD^*) and *Letm1* (**G1**; *Letm1RNAi^HMS^*, **H1**; *Letm1RNAi^GD^*). (**E1**) An intact cyst composed of only 16 cells in an *EndoBRNAi^HMS^* testis, (**F1**) an intact cyst composed of only 32 cells in an *EndoBRNAi^GD^* testis, (**G1**), an intact cyst composed of only 16 cells in a *Letm1RNAi^HMS^* testis, and (**H1**) an intact cyst composed of only 32 cells in a *Letm1RNAi^KK^* testis. Bars: 10 μm.

**Figure 7 ijms-26-09991-f007:**
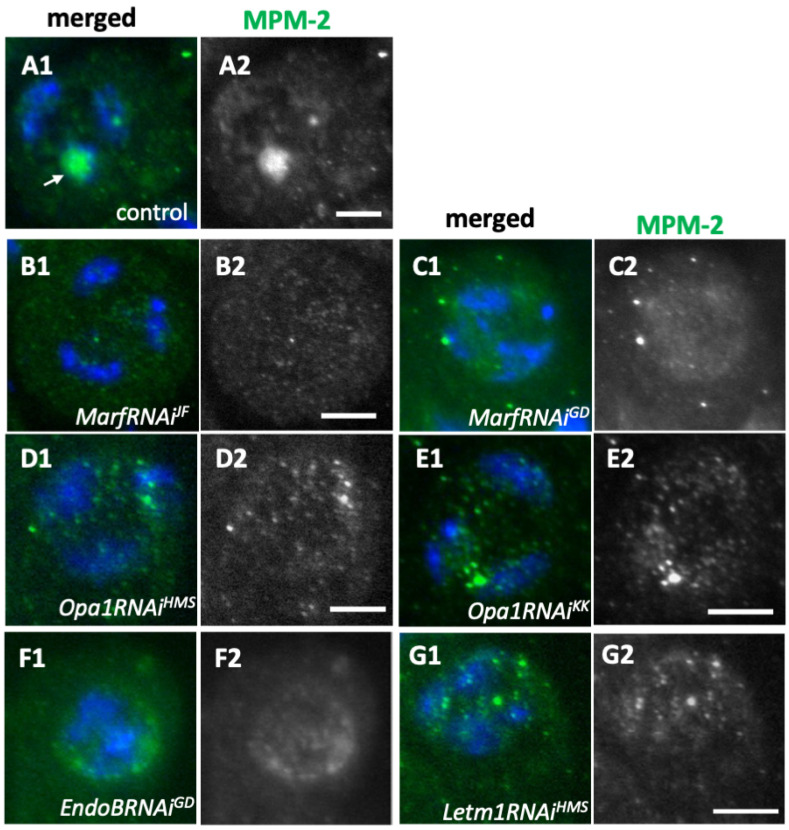
The absence of activated cyclin-dependent kinases in spermatocytes subjected to knockdown of mitochondrial fusion and fission factors. (**A1**–**G1**) Immunostaining of spermatocytes with the MPM2 antibody that recognizes proteins phosphorylated by CDK1 (green in (**A1**–**G1**) and white in (**A2**–**G2**)). (**A1**) In wild-type spermatogonia before the onset of meiosis, a strong anti-MPM2 signal can be observed on the nucleolus (arrow). (**B1**–**G1**) The spermatocytes subjected to knockdown of the mitochondrial fusion factors Marf (**B1**: *MarfRNAi^JF^*, **C1**: *MarfRNAi^GD^*) and Opa1 (**D1**: *Opa1RNAi^HMS^*, **E1**: *Opa1RNAi^KK^*). (**F1**,**G1**) The spermatogonia subjected to knockdown of EndoB (**F1**: *EndoRNAi^GD^*) and Letm1 (**G1**: *Letm1RNAi^HMS^*). Blue in (**A1**–**G1**): DNA. Bars: 10 μm.

**Figure 8 ijms-26-09991-f008:**
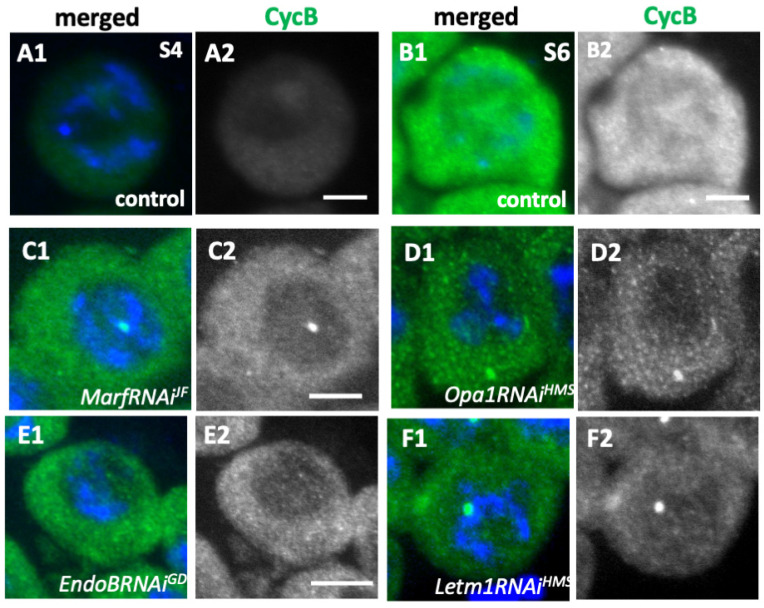
Loss of Cyclin B accumulation in the nuclei of spermatocytes with mitochondrial fusion factors and the morphology proteins knocked down. (**A1**–**F1**)Anti-Cyclin B (CycB) immunostaining of primary spermatocytes (green in (**A1**–**F1**) and white in (**A2**–**F2**)). Blue in (**A1**–**F1**): DNA. (**A1**,**B1**) The immunostaining signals in wild-type spermatocytes in the mid-cell growth phase (the S4 stage) (**A1**), and S6 (**B1**–**F1**). (**C1**–**F1**) The spermatocytes subjected to knockdown of Marf (**C1**: *MarfRNAi^JF^*,) and Opa1 (**D1**: *Opa1RNAi^HMS^*), and those with EndoB (**E1**: *EndoBRNAi^GD^*) and Letm1 (**F1**: *Letm1RNAi^HMS^*) knocked down at the most developed stages. Bars: 10 μm.

**Figure 9 ijms-26-09991-f009:**
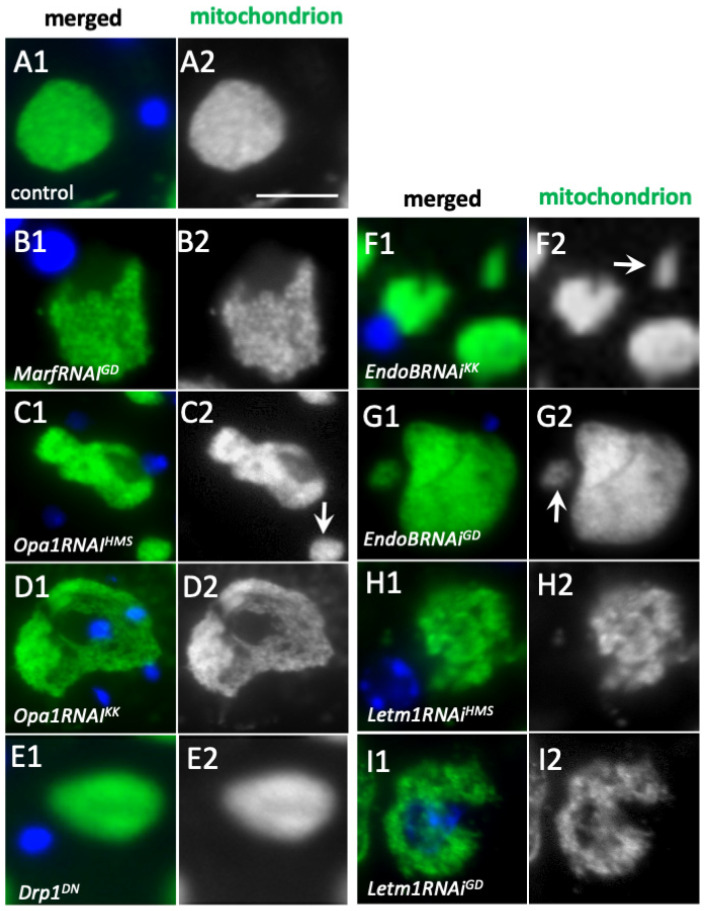
Abnormal formation of Nebenkerns in spermatids derived from spermatocytes subjected to knockdown of mitochondrial fusion factors and morphology proteins, and those from spermatocytes expressing *Drp1^DN^*. (**A1**–**I1**) Immunostaining of spermatids at the onion stage conducted to visualize mitochondria (green in (**A1**–**I1**) and white in (**A2**–**I2**)). Blue in (**A1**–**I1**): DNA staining. (**A1**) Control spermatid. (**B1**–**D1**) Spermatids derived from spermatocytes with Marf (**B1**; *MarfRNAi^GD^*) and Opa1 (**C1**; *Opa1RNAi^HMS^* and **D1**; *Opa1RNAi^KK^*) knocked down. (**E1**) A spermatid from a spermatocyte expressing *Drp1^DN^*. (**F1**–**I1**) Spermatids derived from spermatocytes subjected to knockdown of EndoB (**F1**; *EndoRNAi^JF^* and **G1**; *EndoBRNAi^GD^*) and Letm1 (**H1**: *Letm1RNAi^HMS^* and **I1**: *Letm1RNAi^GD^*). Arrows in (**C2**,**F2**,**G2**): These spermatids possess multiple small Nebenkerns. Bar: 10 µm.

**Table 1 ijms-26-09991-t001:** Frequencies of abnormal spermatid cysts, which resulted from meiotic defects in spermatocytes in which mitochondrial fusion and fission factors and the morphology proteins were depleted.

Knockdown and Dominant Negative exp.	16-Cell Cysts *	17-31-Cell Cysts	32-Cell Cysts	33-63-Cell Cysts	64-Cell Cysts (Normal)
control	0 (0)	0 (0)	0 (0)	0 (0)	68 (100)
*MarfRNAi^JF^*	0 (0)	0 (0)	0 (0)	0 (0)	50 (100)
*MarfRNAi^GD^*	0 (0)	0 (0)	0 (0)	8 (20.0)	32 (80.0)
*Opa1RNAi^HMS^*	3 (3.2)	0 (0)	2 (2.2)	0 (0)	88 (94.6)
*Opa1RNAi^KK^*	97 (84.3)	16 (15.7)	0 (0)	0 (0)	0 (0)
*Drp1RNAi^JF^*	0 (0)	0 (0)	0 (0)	0 (0)	59 (100)
*Drp1^DN^*	0 (0)	0 (0)	0 (0)	30 (58.8)	21 (41.2)
*EndoBRNAi^KK^*	54 (100)	0 (0)	0 (0)	0 (0)	0 (0)
*EndoBRNAi^GD^*	42 (82.4)	0 (0)	9 (17.6)	0 (0)	0 (0)
*Letm1RNAi^HMS^*	40 (100)	0 (0)	0 (0)	0 (0)	0 (0)
*Letm1RNAi^GD^*	42 (84.0)	0 (0)	8 (16.0)	0 (0)	0 (0)

The quantities of intact spermatid cysts (%) observed in the testes with each genotype (*n* > 50 cysts were examined). * Intact spermatid cysts composed of the number of cells indicated there.

**Table 2 ijms-26-09991-t002:** Frequencies of spermatid cysts with abnormal quantities and sizes of nuclei due to abnormalities in meiotic divisions, which were derived from spermatocytes harboring *opa1* depletion.

Knockdown	*n*	Number of Nuclei in a Spermatid (%)							Macro/Micro Nuclei (%)
		Normal	Abnormal						
		1	0	2	3	4	5>	Total	
control	1290	99.5	0.4	0.1	0	0	0	0.5	0
*Opa1RNAi^HMS^*	1713	27.1	7	24.3	15.8	12.3	13.4	73.9	23.5
*Opa1RNAi^KK^*	952	70	29.9	12.2	11.6	4.6	1.5	30	32.4

**Table 3 ijms-26-09991-t003:** A partial rescue of the failure of meiotic division via the knockdown of *Opa1* and *EndoB* via overexpression of Cyclin B (CycB).

Knockdown and Ectopic Expression	16 Cell- Cysts (%) *^1^	17~31 Cell -Cysts	32 Cell -Cysts *^2^	33~63 Cell -Cysts	64 Cell -Cysts *^3^	Total Cysts
*Opa1RNAi^KK^*, *mCherry*	91 (84.3)	16 (14.8)	1 (0.9)	0 (0)	0 (0)	108
*Opa1RNAi^KK^*, *CycB*	67 (51.9)	7 (5.4)	33 (25.6)	4 (3.1)	18 (14.0)	129
*EendoBRNAi^KK^*, *mCherry*	67 (62.6)	26 (24.3)	3 (2.8)	11 (10.3)	0 (0)	107
*EndoBRNAi^KK^*, *CycB*	8 (7.6)	17 (16.2)	11 (10.5)	40 (38.1)	29 (27.6)	105
*CycB*	0 (0)	0 (0)	0 (0)	0 (0)	106 (100)	106

*^1^ The number of spermatid cysts consisting of 16 cells (%), indicating that the cysts were generated from spermatocytes without any meiotic division. *^2^ The number of spermatid cysts consisting of 32 cells (%), indicating that the cysts were generated from spermatocytes via only one of either meiotic division. *^3^ The number of spermatid cysts consisting of 64 cells (%), indicating that the cysts were generated from spermatocytes via meiotic division I and II.

## Data Availability

The datasets generated and/or analyzed in the current study are available from the corresponding author upon reasonable request.

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
