# Peer review of "Marf- and Opa1-Dependent Formation of Mitochondrial Network Structure Is Required for Cell Growth and Subsequent Meiosis in Drosophila Males"

_ijms, 2025, doi:10.3390/ijms26209991_

Round 1
Reviewer 1 Report
Comments and Suggestions for Authors
The study reports that during Drosophila spermatogenesis an interconnected network of elongated mitochondria forms before meiosis and is maintained during meiotic divisions, relying on microtubules and F-actin. Knockdown of mitochondrial fusion factors (Opa1 and Marf) and morphology proteins (Letm1 and EndoB) inhibited cell growth, disrupted meiosis, prevented proper Cyclin B accumulation and Cdk1 activation, and led to spermatids with multiple smaller nuclei and abnormal Nebenkerns. While the study provides potentially interesting insights into mitochondrial dynamics during male germline cell division, the manuscript needs substantial restructuring. Due to the large volume of detailed information, a reorganization of the text is necessary to improve clarity for the reader and the overall presentation of results and discussion.
Introduction
1- Line 77 “However, it remains unclear…”, Line 117 “The fine morphology…” and Line 50 “However, no direct evidence has been…”. Mentions of the lack of knowledge regarding mitochondrial morphological changes appear in more than one section, which may create repetition. Focusing this emphasis at the end of the introduction, alongside the study objectives, could make the text clearer and more concise.
2- Line 80-105. “The morphology of mitochondria…”. The paragraph on Drosophila spermatogenesis is quite long, which can make it dense for an introduction. The meticulous enumeration of the cellular stages, while accurate, may distract from the focus and could be streamlined into broader blocks that highlight only the essential aspects of mitochondrial changes. Additionally, connecting this description earlier to the central point of the study, the gap in knowledge regarding how these morphological changes affect meiosis, would make the text more direct and aligned with the study objectives.
3- Line 43-50. “When oxidative damages…”, Line 53-60. “The studies on the molecular mechanisms…” and Line 106-111. “In Drosophila, two orthologues…”
The concept of mitochondrial fusion and fission is repeated in several sections of the introduction. This repetition contributes to a denser presentation and makes the introduction as a mini review, rather than leading directly to the study’s gap and objectives.
Results
4- In general, the use of A, A′, A″ to indicate different versions of the same figure may be confusing for readers. Using a sequential labeling system such as A1, A2, A3 for all figures would make them clearer and easier to reference in the text.
5- In general, the figure legends are very long and often mixing description with interpretation and methodological details. It is recommended to revise all legends to focus specifically on what is shown in the panels and make them more concise.
6- Many of the subtitles in the Results section are very long and detailed, some have almost three lines. It is recommended to review all subtitles and consider shortening them to make the section more concise and easier for the reader. For example: Line 132 – Subtitle 2.1 Could be shortened to ‘Dynamics of mitochondrial morphology in Drosophila spermatogenesis’ to make it more concise and easier to read.
7- Line 134–136. This section includes methodological details (immunostaining and confocal microscopy) that are more appropriate for the Materials and Methods section. In the Results, the focus should be directly on the observations of mitochondrial morphology. In general, similar issues appear throughout all sections of the Results, where methodological descriptions are mixed with observations, making the narrative dense and harder to follow (Topics 2.1–2.9).
8- Many methodological details are currently mixed within the Results, which makes the narrative dense and harder to follow. Given the complexity and multiple steps of the experimental procedures, the authors could greatly benefit from including schematic diagrams or graphical summaries in the Methods section to illustrate the experimental design and workflow. This would enhance clarity and make the manuscript more accessible to readers.
9- Line 337. Figure 5. Comparisons against control satisfy the main objective of the study. However, it could be informative to compare all knockdown groups with each other to assess their relative effects on spermatocyte diameter.
Discussion
10- Line 563-564. The sentence ‘We have performed immunostaining to identify mitochondria in meiotic cells…’ is disconnected from the rest of the paragraph, which focuses on interpreting results and comparing with previous studies. This methodological detail is unnecessary in the Discussion and could be removed. The paragraph would benefit from being rewritten to improve flow, focusing on interpretation and synthesis rather than experimental procedures.
11- Line 567-571. The sentences describing observations from Fuller (1993), including the repeated references and detailed measurements of mitochondria, appear disconnected from the rest of the Discussion. It is recommended to either condense these statements into a single reference noting consistency with previous findings or integrate them into the discussion of biological significance.
12- Line 606. Replace ‘Aldridge and colleagues reported’ with ‘Aldridge et al. (2007) reported’ to follow standard scientific citation.
13- Line 621-630. The paragraph mixes observations from the present study with references to mammalian cells in a confusing way. It would be clearer to first present the results obtained in this study and then compare them with what is known in mammalian cells. As the hypothesis about F-actin is not new and not supported by data here, it should be referenced with studies that have demonstrated its role in mitochondrial dynamics.
14- Line 652-654. The sentences speculating that ROS may influence the progression of the growth phase in Drosophila are not supported by any evidence presented in this study. Although observations in mammalian cells are cited, no data is provided that link these effects to Drosophila meiosis. We therefore suggest removing this paragraph, as it lacks supporting evidence and relevance to the findings.
15- Line 673-688 This paragraph in the Discussion largely reiterates well-established roles of microtubules and F-actin in mitochondrial transport. It does not provide new experimental evidence from the current study regarding F-actin. The section could be made more concise by focusing on the authors’ own findings, avoiding unnecessary repetition of literature, and clearly distinguishing between observations and speculation.
16- Line 725-732 The conclusion paragraph at the end of the discussion is uninformative. It does not clearly state the main findings or their significance, focusing mainly on limitations.
Methods
17- Line 735-760 – Subtitle 4.1 Contain a very long list of Drosophila stocks with full details, which makes the text dense and hard to read. It may be more effective to use the abbreviations in the main text and provide a detailed table in the supplementary materials with full stock names, codes, origins, and references.
18- Lines 836–840 – Subtitle 4.7 Mention that Student’s t-test was applied as described in a previous study. It would be helpful to briefly justify why this test is appropriate for the current datasets, rather than citing prior work, to clarify the methodological rationale.
Author Response
The authors appreciate Reviewer 1 for her/his careful reading and for providing thoughtful comments. To improve our manuscript, we carefully reviewed all of the reviewers’ comments and responded to each one individually. Additionally, considering reviewer 1’s assessment of the English quality, we contacted MDPI's English proofreading service and revised the language as indicated (English editing ID: english-101412).
Introduction 
1- Line 77 “However, it remains unclear…”, Line 117 “The fine morphology…” and Line 50 “However, no direct evidence has been…”. Mentions of the lack of knowledge regarding mitochondrial morphological changes appear in more than one section, which may create repetition. Focusing this emphasis at the end of the introduction, alongside the study objectives, could make the text clearer and more concise.
In accordance with Reviewer 1s' suggestion, we consolidated the repeated statements regarding the lack of sufficient knowledge on the mitochondrial morphological alteration during early spermatogenesis into a single key issue at the end of the introduction (lines 82-84). We shortened the Introduction by 33 lines (from 130 to 97 lines) in total to clarify the research objectives.
2- Line 80-105. “The morphology of mitochondria…”. The paragraph on Drosophila spermatogenesis is quite long, which can make it dense for an introduction. The meticulous enumeration of the cellular stages, while accurate, may distract from the focus and could be streamlined into broader blocks that highlight only the essential aspects of mitochondrial changes. Additionally, connecting this description earlier to the central point of the study, the gap in knowledge regarding how these morphological changes affect meiosis, would make the text more direct and aligned with the study objectives.
Taking the comment into account, we omitted a detailed enumeration at the cellular level, instead describing only the key points necessary to understand the novel findings of this study. To clarify the research objectives, we shortened the paragraph on Drosophila spermatogenesis in the introduction to the minimum necessary for the readers outside the field to understand.
3- Line 43-50. “When oxidative damages…”, Line 53-60. “The studies on the molecular mechanisms…” and Line 106-111. “In Drosophila, two orthologues…”
The concept of mitochondrial fusion and fission is repeated in several sections of the introduction. This repetition contributes to a denser presentation and makes the introduction as a mini review, rather than leading directly to the study’s gap and objectives.
Among the points where the reviewer noted that the concepts of mitochondrial fusion and fission were repeatedly mentioned, we have deleted some descriptions that are not directly related to the results of this study. The introduction has been shortened to 75% of the original manuscript.
Results
4- In general, the use of A, A′, A″ to indicate different versions of the same figure may be confusing for readers. Using a sequential labeling system such as A1, A2, A3 for all figures would make them clearer and easier to reference in the text.
As requested by the reviewer, we revised panel labels as A1, A2, A3 for all figures presenting images.
5- In general, the figure legends are very long and often mixing description with interpretation and methodological details. It is recommended to revise all legends to focus specifically on what is shown in the panels and make them more concise.
Following the reviewer’s request, we removed descriptions containing interpretation and methodological details from all nine figure legends.
6- Many of the subtitles in the Results section are very long and detailed, some have almost three lines. review all subtitles and consider shortening them to make the section more concise and easier for the reader. For example: Line 132 – Subtitle 2.1 Could be shortened to ‘Dynamics of mitochondrial morphology in Drosophila spermatogenesis’ to make it more concise and easier to read.
Following the reviewer's recommendation, we revised most of the subtitles (all 9 except 2.3 and 2.8), including Subtitle 2.1, to make them shorter.
7- Line 134–136. This section includes methodological details (immunostaining and confocal microscopy) that are more appropriate for the Materials and Methods section. In the Results, the focus should be directly on the observations of mitochondrial morphology. In general, similar issues appear throughout all sections of the Results, where methodological descriptions are mixed with observations, making the narrative dense and harder to follow (Topics 2.1–2.9).
Following Reviewer 1's request, we removed redundant methodology descriptions (16 items) from the Results section and summarized them in the Materials and Methods section.
8- Many methodological details are currently mixed within the Results, which makes the narrative dense and harder to follow. Given the complexity and multiple steps of the experimental procedures, the authors could greatly benefit from including schematic diagrams or graphical summaries in the Methods section to illustrate the experimental design and workflow. This would enhance clarity and make the manuscript more accessible to readers.
In accordance with Reviewer 1's recommendation, we removed redundant methodology descriptions from the Results section as much as possible and consolidated them into the Materials and Methods section. Furthermore, following the reviewer’s recommendation, we added schematic diagrams to illustrate the experimental workflow for peer review only (Figure for reviewing).
9- Line 337. Figure 5. Comparisons against control satisfy the main objective of the study. However, it could be informative to compare all knockdown groups with each other to assess their relative effects on spermatocyte diameter.
To compare the knockdown effects between fusion factors themselves or between fusion factors and morphogenetic factors, and thereby compare their relative involvement in cell growth, further additional experiments would be necessary. In this study, we aim to present simply the results regarding whether the individual mitochondrial fusion and morphogenetic factors investigated here affect spermatocyte growth. We appreciate the reviewers' valuable suggestions for future experimental plans.
Discussion
10- Line 563-564. The sentence ‘We have performed immunostaining to identify mitochondria in meiotic cells…’ is disconnected from the rest of the paragraph, which focuses on interpreting results and comparing with previous studies. This methodological detail is unnecessary in the Discussion and could be removed. The paragraph would benefit from being rewritten to improve flow, focusing on interpretation and synthesis rather than experimental procedures.
As per the reviewer’s request, we removed the sentence that were in lines 563-564 of the original manuscript.
11- Line 567-571. The sentences describing observations from Fuller (1993), including the repeated references and detailed measurements of mitochondria, appear disconnected from the rest of the Discussion. It is recommended to either condense these statements into a single reference noting consistency with previous findings or integrate them into the discussion of biological significance.
Considering this reviewer’s comments, we revised this part to make the flow of the discussion clearer, removing repetitive references and detailed descriptions on mitochondrial measurements. We consolidated the previous findings from several references to state consistency with our findings and claim the novelty of our findings (lines 503- 512).
12- Line 606. Replace ‘Aldridge and colleagues reported’ with ‘Aldridge et al. (2007) reported’ to follow standard scientific citation.
We revised the phrase according to the reviewer’s request (line 545).
13- Line 621-630. The paragraph mixes observations from the present study with references to mammalian cells in a confusing way. It would be clearer to first present the results obtained in this study and then compare them with what is known in mammalian cells. As the hypothesis about F-actin is not new and not supported by data here, it should be referenced with studies that have demonstrated its role in mitochondrial dynamics.
Considering the comment, we revised the structure of Section 3.2 of the Discussion. We first summarize our key results and compare them with known findings in Drosophila spermatogenesis, followed by previous findings in mammalian cells. Regarding the hypothesis on the role of F-actin in forming the network structures, the explanation was too concise, leading to misunderstanding. Therefore, based on the data from this study, we reorganized the explanation and added additional text and supporting literature (lines 561-587).
14- Line 652-654. The sentences speculating that ROS may influence the progression of the growth phase in Drosophila are not supported by any evidence presented in this study. Although observations in mammalian cells are cited, no data is provided that link these effects to Drosophila meiosis. We therefore suggest removing this paragraph, as it lacks supporting evidence and relevance to the findings.
Following the reviewer's suggestion, we removed the sentences that were present in lines 652-654 of the original manuscript.
15- Line 673-688 This paragraph in the Discussion largely reiterates well-established roles of microtubules and F-actin in mitochondrial transport. It does not provide new experimental evidence from the current study regarding F-actin. The section could be made more concise by focusing on the authors’ own findings, avoiding unnecessary repetition of literature, and clearly distinguishing between observations and speculation.
In response to the reviewer's comment, we have focused on the novel findings from this study. We have avoided unnecessary repetition and distinguished between the observed results and the speculation that can be drawn from them, ensuring the points we wish to argue in this section (lines 627-642).
16- Line 725-732 The conclusion paragraph at the end of the discussion is uninformative. It does not clearly state the main findings or their significance, focusing mainly on limitations.
Considering the reviewer’s comment, we added a conclusion section 3.6. at the end of the discussion section to highlight the main new findings and their biological significance of this study (lines 679-691).
Methods
17- Line 735-760 – Subtitle 4.1 Contain a very long list of Drosophila stocks with full details, which makes the text dense and hard to read. It may be more effective to use the abbreviations in the main text and provide a detailed table in the supplementary materials with full stock names, codes, origins, and references.
Following the reviewers' suggestions, we used abbreviations for Drosophila stocks in the main text and included the supplementary table (Table S2) in the supplementary materials listing stock names, stock numbers, origins, and references.
18- Lines 836–840 – Subtitle 4.7 Mention that Student’s t-test was applied as described in a previous study. It would be helpful to briefly justify why this test is appropriate for the current datasets, rather than citing prior work, to clarify the methodological rationale.
To confirm that the reduction in cell size observed in spermatocytes with knockdown (for each RNAi line) was statistically significant compared to the control, we performed 1:1 comparisons between the control and each knockdown using a Student’s t-test. Before then, we initially performed an F-test to determine whether the variances were equal or unequal. Student’s t-test was selected to perform when the value was greater than 0.05 (with equal variance), whereas Welch’s t-test was performed when the value was less than 0.05 (unequal variance). We did not compare the knockdowns with each other this time, as we believe that the Student's t-test is appropriate for this purpose. According to the comment, we added this explanation in the Materials and Methods section 4.7. (Lines 781–786).

Reviewer 2 Report
Comments and Suggestions for Authors
This study has generated unique data on formation and dynamics of mitochondrial networks within male germ cells, and how mitochondrial fusion factors and morphology proteins regulate cell growth and progression through meiosis. Data are well presented and described. I have a few comments for the authors to address:
Line 275: "...confirmed that each mRNA was depleted by less than 10% of controls (Fig. S3)". This implies only a 10% reduction whereas the reduction is approximetely 90%. Please reword this statement. I also noted that the endoB reduction was not quite as complete as the other genes.
Line 283 refers to the Drp1DN allele but the data were not present in Figure 4. A similar stement is made on line 299 but again no data were related to this allele in Fig S4B.
Figure S1 appears to have panels cropped off to the right and panel C'' is very pixellated.
The phenotypes observed by super resolution microscopy in Fig S4 seem qualitatively different when comparing the Opa1RNAi and EndoBRNAi. Could the authors please comment.
Section 2.4 reports ATP amounts associatd with the various alleles but the data are not present in any figures or tables.
Line 379 describes that Opa1 knockdown cysts frquently only have 16 cells. Could the authors also please comment on the morphology of these cells.
Line 383 comments upon twe mutant phenotypes. Could a reference to this please be added.
Line 471. I believe the reference to Fig 8A' should be Fig 8B'.
Figure 8. The Marf and EndoB knockdowns appear to have higher levels of CycB at S4 compared to the control. Could the authors please comment upon this.
Author Response
The authors appreciate Reviewer 2 for her/his careful reading and thoughtful comments. To improve our manuscript, we carefully reviewed all of the reviewers’ comments and responded to each one individually. Additionally, considering reviewer 1’s assessment of the English quality of our manuscript, we contacted MDPI's English proofreading service and revised the language as indicated (English editing ID: english-101412).
1.Line 275: "...confirmed that each mRNA was depleted by less than 10% of controls (Fig. S3)". This implies only a 10% reduction whereas the reduction is approximetely 90%. Please reword this statement. I also noted that the endoB reduction was not quite as complete as the other genes.
It was an error in English grammar. We revised the sentence as follows. ” Ectopic expression of dsRNAs against Opa1, Marf, Drp1, and EndoB mRNAs in the spermatocytes decreased the relative quantity of mRNAs to less than 10% of the control group.” (line 229 in the revised manuscript).
2. Line 283 refers to the Drp1DN allele but the data were not present in Figure 4. A similar stement is made on line 299 but again no data were related to this allele in Fig S4B.
In cells with Drp1 knockdown or a Drp1 dominant-negative mutant, mitochondrial morphology showed little difference from the control group. To prevent the image data in Figure 4 from becoming overly complex, we have provided a simple description that there was no difference from the control in the text (lines 237-240).
3. Figure S1 appears to have panels cropped off to the right and panel C'' is very pixellated.
These were editing errors and an oversight by this corresponding author. We have corrected those errors appropriately (Figure S1). We appreciate the reviewer's careful review.
4. The phenotypes observed by super resolution microscopy in Fig S4 seem qualitatively different when comparing the Opa1RNAi and EndoBRNAi. Could the authors please comment.
As noted in the reviewer's comments, the mitochondria in mature spermatocytes from EndoBRNAi (Fig. S4C) appeared slightly longer compared to those from Opa1RNAi (Fig. S4A). While Opa1 is a GTPase essential for mitochondrial fusion, EndoB is a morphology protein required for altering the inner mitochondrial membrane morphology. Therefore, we agree that knocking down both proteins prevents the formation of an elongated mitochondrial network, but the mitochondrial shapes observed are qualitatively different. However, this study focuses on the role of fusion factors during the early stages of spermatogenesis. To keep the study focused, we did not explore the differences with EndoB in detail in this study.
5. Section 2.4 reports ATP amounts associatd with the various alleles but the data are not present in any figures or tables.
Considering the reviewers' comments that the data on ATP amounts associated with the various alleles were not presented in any figures or tables, we compiled them into Table S4, in accordance with the format used in our previous paper (Azuma et al., 2019)..
6. Line 379 describes that Opa1 knockdown cysts frquently only have 16 cells. Could the authors also please comment on the morphology of these cells.
According to the reviewer’s comment, we added a phrase underlined at the end of this sentence: “Consistently, all intact spermatid cysts from spermatocytes with knockdown of another fusion factor, Opa1 (Opa1RNAiHMS), contained only 16 cells, each with a single nucleus that was larger than in controls and a Nebenkern.“ (lines 332-333). This phenotype closely resembles that of twe, which is essential for the initiation of meiosis. Therefore, it is highly likely that spermatocytes with the Opa1 knockdown also differentiated into spermatids without undergoing meiosis.
7. Line 383 comments upon twe mutant phenotypes. Could a reference to this please be added.
Accordingly, we added the relevant references (line 336).
8. Line 471. I believe the reference to Fig 8A' should be Fig 8B'.
We revised the mistake accordingly (line 421).
9. Figure 8. The Marf and EndoB knockdowns appear to have higher levels of CycB at S4 compared to the control. Could the authors please comment upon this.
Fig. 8C and E images represent the anti-CycB immunostaining images of the knockdown spermatocytes at S6 but not S4 stage. These knockdown cells exhibit CycB expression that is slightly higher than the basal level in controls at S4. However, it is significantly lower than the level observed in normal mature spermatocytes at S6.

Round 2
Reviewer 2 Report
Comments and Suggestions for Authors
I am satisfied by the revisions to the manuscript.